# Chemical Defoliant Promotes Leaf Abscission by Altering ROS Metabolism and Photosynthetic Efficiency in *Gossypium hirsutum*

**DOI:** 10.3390/ijms21082738

**Published:** 2020-04-15

**Authors:** Dingsha Jin, Xiangru Wang, Yanchao Xu, Huiping Gui, Hengheng Zhang, Qiang Dong, Ripon Kumar Sikder, Guozheng Yang, Meizhen Song

**Affiliations:** 1State Key Laboratory of Cotton Biology, Institute of Cotton Research, Chinese Academy of Agricultural Sciences, Anyang 455000, China; jindingsha@163.com (D.J.); wxr_z4317@163.com (X.W.); xuyanchao2016@163.com (Y.X.); guihuiping@caas.cn (H.G.); zhanghengheng@caas.cn (H.Z.); dongqiang@caas.cn (Q.D.); 2017Y90100144@caas.cn (R.K.S.); 2MOA Key Laboratory of Crop Eco-physiology and Farming system in the Middle Reaches of Yangtze River, College of Plant Science and Technology, Huazhong Agricultural University, Wuhan 430000, China; 3School of Agricultural Sciences, Zhengzhou University, Zhengzhou 450001, China

**Keywords:** cotton, leaf abscission, ROS, photosynthesis, carbohydrate, RNA-seq

## Abstract

Chemical defoliation is an important part of cotton mechanical harvesting, which can effectively reduce the impurity content. Thidiazuron (TDZ) is the most used chemical defoliant on cotton. To better clarify the mechanism of TDZ promoting cotton leaf abscission, a greenhouse experiment was conducted on two cotton cultivars (CRI 12 and CRI 49) by using 100 mg L^−1^ TDZ at the eight-true-leaf stage. Results showed that TDZ significantly promoted the formation of leaf abscission zone and leaf abscission. Although the antioxidant enzyme activities were improved, the reactive oxygen species and malondialdehyde (MDA) contents of TDZ increased significantly compared with CK (water). The photosynthesis system was destroyed as net photosynthesis (Pn), transpiration rate (Tr), and stomatal conductance (Gs) decreased dramatically by TDZ. Furthermore, comparative RNA-seq analysis of the leaves showed that all of the photosynthetic related genes were downregulated and the oxidation-reduction process participated in leaf shedding caused by TDZ. Consequently, a hypothesis involving possible cross-talk between ROS metabolism and photosynthesis jointly regulating cotton leaf abscission is proposed. Our findings not only provide important insights into leaf shedding-associated changes induced by TDZ in cotton, but also highlight the possibility that the ROS and photosynthesis may play a critical role in the organ shedding process in other crops.

## 1. Introduction

Organ shedding is a common phenomenon in plants, which is of great significance to the growth and development during the lifespan of a plant, especially the yield of crops [1]. Abscission ensures that plants can shed organs when they are no longer needed, such as flowers, fruits, and senescent leaves [2] and also as a means of escaping infected or damaged organs [3]. Once abscission is triggered, the abscission zone (AZ), which consists of small and dense cells and interconnected by plasmodesmata, occurs, and then hydrolytic enzymes dissolve the abscission zone’s middle lamella, resulting in organ abscission [2,4].

Previous studies have found that ethylene accelerates organ abscission, while auxin inhibits this process [5,6,7,8,9]. Meanwhile, researchers have paid attention to the fact that multiple cell wall-degrading enzymes are activated to dissolve the cell wall and middle lamella of abscission zone, including cellulases, pecticlyases, polygalactosidases [10,11,12]. Some research has found that hydrogen peroxide (H_2_O_2_) acts downstream from ethylene and plays a role in the cell-wall degradation process in abscission signaling [13,14].

Reactive oxygen species (ROS), including H_2_O_2_, superoxide, singlet oxygen, and the hydroxyl radical, are produced in response to environmental stress [13]. Environmental stresses (such as drought, cold stress, salt stress, and pathogen attack) are often accompanied by leaf abscission [15,16,17], but the relationship between stress and abscission is rarely studied. Excessive ROS can damage cellular components, including lipids, protein, and nucleic acids [18]. ROS produced at the abscission zone site play an important role in regulating leaf abscission under drought stress in cassava (*Manihot esculenta* Crantz) [19]. Recent studies have found that ROS do not work in the fruit abscission zone of olives (*Olea europaea* L.), but only alter oxidative stress in the abscission zone of leaves and then mediate abscission induced by ethephon [20]. Previous studies have shown that insufficient carbohydrate accumulation and distribution can lead to flower and fruit abscission. For example, carbohydrate stress can induce longan (*Dimocarpus longan* Lour.) fruit abscission, which may be mediated by ROS [21]. Carbohydrate content may be a biochemical signal controlling Citrus (*Citrus unshiu* [Mak.] Marc.) fruits abscission [22]. Carbon starvation, caused by shade, could increase flower drop rates [23]. However, there are few studies on the relationship between leaf abscission and carbon stress. While there is a mass of physiological and molecular knowledge uncovering how the abscission zone of leaves responds to a variety of abscission, much less is known about the leaf.

Cotton (*Gossypium hirsutum* L.) is an important economic crop worldwide, which provides fiber materials for the textile industry [24]. Mechanized harvesting is the trend of cotton production in China since it is the key measure to improve harvesting efficiency and solve the problem of labor shortage [25,26]. Applying chemical defoliants before harvesting can promote the shedding of cotton leaves as well as promote boll opening and thus effectively reduce the content of impurities in raw cotton and enhance the harvest efficiency, especially for the mechanically harvested cotton [25,27]. Thidiazuron (a synthetic cytokinin-like molecule, TDZ) is the main component of cotton chemical defoliants that widely used in most cotton-producing countries currently [28]. However, the mechanism of TDZ inducing cotton leaf shedding is not completely clear. In addition, there are still some problems in the practical use of defoliant, for example, the leaves dehydrated but not shed, and the effect of defoliants being tightly associated with the application schedule, dosage, and air temperature. Early studies showed that adenine cytokinins could induce cotton leaves abscission by increasing the content of ethylene [29,30]. Botton et al. also reported that cytokinin (ben-zyladenine, BA), could cause fruits abscission by crosstalk signaling pathways, mainly involving sugars, ROS, ABA, and ethylene [31]. Recent studies on TDZ elucidated that TDZ could regulate cotton defoliation by increasing cell wall-degrading enzyme activity and ethylene content [25] and the crosstalk between the cytokinin and ethylene signaling pathway commonly regulates cotton defoliation in response to TDZ [9]. Whether the exogenous TDZ induces leaf abscission related to ROS and carbohydrate change remains uncertain. Therefore, we conducted a greenhouse experiment to investigate the cotton leaf abscission induced by TDZ. Changes in leaf structure and physiological indexes (ROS and photosynthesis) were tested, while the transcriptomes of the leaves of CRI 12 at 48 h following TDZ treatment were examined in this study.

## 2. Results

### 2.1. Morphology and Anatomical Features of Cotton Seedlings during Leaf Shedding

Cotton leaves sprayed with 100 mg L^−1^ TDZ appeared to be premature defoliated (Figure 1). Seedling leaves of CRI 12 and CRI 49 showed similar phenotypic characteristics under the treatment of TDZ, e.g., white spots appearance, water loss, as well as downward roll-up of leaves. A tear formed at the abscission zone (AZ) and purple spots appeared on both the leaves and stem of cotton seedlings (Figure 1a–c). However, no obvious change of phenotypic characteristics was observed in CK.

As shown in Figure 1d,e, the AZ occurred 1 d after treatment and the abscission zone formation rate (AZR) sharply increased to 90% at the time point of 4 d in CRI 12, one day earlier and 30 percent higher AZR compared to CRI 49. Accordingly, the leaf abscission was observed firstly in CRI 12 at the 2-d point, and the abscission rate (ABR) reached 80% at 7 d after treatment. The leaf of CRI 49 began to shed 3 d after treatment and the final ABR was only 20%, 60 percent lower than that of CRI 12. All these results above suggest that CRI 12 is more sensitive to TDZ compared with CRI 49.

Disassembly of cell walls in the leaf should lead to altered anatomical features. The leaf epidermis was examined under scanning electron microscopy in order to elucidate the anatomical alterations. Upper epidermis cells of the leaf were broken after spraying TDZ. Stoma was severely devastated (Figure 1f). The closely arranged, well-organized, and irregular cells were observed on the leaf epidermis of control plants; cells of the upper epidermis became differentiated and damaged by TDZ (Figure 1f). The treated cells appeared to be pulled together into clumps and disorganized compared to the control (Figure 1f).

### 2.2. ROS Homeostasis during Abscission

ROS homeostasis system was investigated to illustrate the early response of TDZ-triggered leaf abscission. Our result showed ROS balance was significantly destroyed by TDZ. H_2_O_2_ content prominently augmented in CRI 12 and CRI 49 at 48 h and was significantly decreased in CRI 49 at 24 h (Figure 2b). The superoxide anion free radical (O_2_^−^) production rate exhibited a similar trend with H_2_O_2_, nevertheless O_2_^−^ content showed no significant change for CRI 12 at 24 h (Figure 2c). Additionally, MDA (malondialdehyde) content of TDZ treatment was markedly higher than that of CK at 24 and 48 h for both CRI 12 and CRI 49 (Figure 2a). Although CAT (catalase) activity was significantly lower in TDZ, the POD (peroxidase) and SOD (superoxide dismutase) were significantly improved (except for SOD at 48 h in CRI 12; Figure 2d–f).

### 2.3. Cotton Photosynthetic System during Abscission

The photosynthetic system of cotton leaves suffered destructive effects under TDZ treatment. Pn rapidly decreased in both cultivars, which may give rise to a high defoliation rate of leaves (Figure 3a). Meanwhile, the change of Tr and Gs showed the same trend as Pn (Figure 3b,d). The intercellular carbon dioxide concentration (Ci) was quite stable and existed unchanged within 12 h after treatment; after that, it significantly increased (Figure 3c). The damage of the photosynthetic system was confirmed with further observation of photosynthetic pigments. Negative effects of TDZ on chlorophyll a (Chla), chlorophyll b (Chlb), carotenoid (Cxc), total chlorophyll (Total Chl), and Chla/Chlb at 24 h were observed. Compared with CK, Chla, Chlb, carotenoid, and total Chl of TDZ treatment increased significantly (Figure 3e–h), whereas Chla/Chlb showed a significant decrease at 48 h (Figure 3i). Anthocyanin content increased remarkably relative to CK at 48 h after treatment for both cultivars (Figure 3j).

### 2.4. Carbohydrate Contents in Leaf during Abscission

The carbohydrate metabolism of the leaves was also investigated to understand the early response to TDZ during abscission. Soluble sugar (SS), sucrose, and starch contents of TDZ were significantly lower than that of CK (Figure 4a,b,d). SS/FAA also exhibited a similar trend with soluble sugar, sucrose, and starch contents at 48 h after treatments (Figure 4f). The fructose contents showed a different trend between different stages and varieties. For CRI 49, fructose content was decreased at 24 h but increased at 48 h by TDZ (Figure 4e). For CRI 12, fructose content was only decreased by TDZ at 24 h. There was no significant difference in free amino acid (FAA) content at an early stage, but the FAA content of TDZ was significantly higher at 48 h (Figure 4c).

### 2.5. ROS Metabolism, Carbohydrate Metabolism and Photosynthesis Process Involved in Early Response under TDZ Treatment

Given the multifaceted differences between the control plants and treated plants, Poisson correlation coefficients were calculated to investigate the biological consequences of TDZ treatment (Figure 5). The cluster analysis of all the indicators could be divided into five categories (Figure 5). According to Cluster I, there were significant positive correlations among AZR, ABR, osmotic stress-related indexes, and photosynthetic pigment content (Figure 5). AZR and ABR were negatively related to photosynthetic indexes and carbohydrate contents. Among 13 significantly correlated indexes, 10 indexes were from 48 h, suggesting that 48 h after treatment may be the key time point for TDZ function. Cluster II-V contains ROS, photosynthetic pigment, and carbohydrate metabolism-related indexes, suggested the potential correlation among three biological processes. The result of two difference testing (physical testing and instrument testing) is consistent with each other, suggested the data we measured are reliable.

### 2.6. Transcriptome Analysis

The phenotypic and physiological data of 48 h have significantly changed and have significant positive correlations with leaf abscission rate. Given the phenotypic and physiological data analyses, we conducted the RNA-seq experiment at 48 h of TDZ treatment on an Illumina Novaseq™ 6000 at the (LC Sciences, USA), in order to identify abscission signal. After the removal of low-quality reads, an average of 4.7 × 10^7^ clean reads were obtained for sample, and the total length of the clean reads reached 2.8 × 10^8^ nt. For each sample, 87% of the clean reads were mapped to the cotton reference transcriptome (additional file: Appendix A). The results showed that 17,865 genes were differentially expressed by T48LVSC48L comparison (DEG, |log2 fold change (FC)| > 1 and *p* < 0.05) and 6844 and 11,021 genes were respectively up- and downregulated. According to the Gene Ontology (GO) and Kyoto Encyclopaedia of Genes and Genomes (KEGG) pathway analyses, the DEGs that changed significantly primarily included pigment (chlorophyll) metabolism, oxidation-reduction process (especially, glutathione metabolism), photosynthesis-related genes, carbon fixation in photosynthetic organisms, plant hormone signal transduction, response to biotic and abiotic stresses, MAPK signaling pathway and plant–pathogen interaction. We found 127 photosynthesis related genes; among them, only one gene was upregulated. Over 99.2% genes were downregulated, indicating that most genes of photosynthesis were largely affected at 48 h after the TDZ treatment. All photosynthesis–antenna protein-related genes were downregulated. The expression of genes related to carbon fixation and alternative oxidase was repressed, while the expression of genes associated with PSI and PSII activities, cytochrome b6/f complex, photosynthesis electron transport chain of mitochondria and F-type ATPase was repressed (Figure 6 and Appendix A).

## 3. Discussion

Leaf abscission, occurring along a fracture plane called the abscission zone, can be triggered not only by internal development cues and but also external environment changes. TDZ is a kind of chemical defoliation agent that significantly induces leaf shedding and the formation of the abscission zone. However, the mechanism of TDZ-triggered leaf abscission is little known. According to the similar phenotypical and physiological characteristics of cotton seedling leaves under abiotic stress and TDZ treatment, the biological process of TDZ-triggered leaf abscission response appears to be similar to the abiotic stress-triggered leaf abscission [19,32]. Our research provides evidence to verify this hypothesis. In addition, our study also focuses on the early response of cotton seedling leaves under TDZ treatment. In addition, the results suggest that ROS metabolism, photosynthesis process, and carbohydrate metabolism are involved in the early response to TDZ. What is more, our results suggest the potential crosstalk among those biological processes regulate leaf abscission.

### 3.1. Phenotypic Characteristic Damage under TDZ Treatment

Cotton leaf dropped from the parent plant with TDZ treatment. Therefore, research on leaf abscission in response to TDZ at the phenotypic change is highly important. Previously, TDZ accelerates cotton leaf abscission, and the anatomical features of AZ are also altered [9,25]. Our results showed that cotton leaf rapidly abscises and the formation of AZ occurs under TDZ treatment, which is consistent with previous studies. Previous studies have focused on anatomical features of AZ. However, the anatomical features of leaves under TDZ treatment remained unclear at the moment. Thus, anatomical features of cotton leaf are observed in this study, and we found that leaf cells and stomata were damaged under TDZ treatment, suggesting there probably exists a great positive correlation between leaf abscission and stoma damage. Stomata formed by a pair of guard cells are the channels through which plants exchange photosynthetic CO_2_ gas and control transpiration, and stomata play an extremely important role in the life of plants [33,34]. Some research has found that ROS accumulation in guard cells could promote stomatal closure [35] and excessive ROS accumulation causes the apoptosis of leaf cells [36]. We established the mechanism by which the leaf epidermis structure was destroyed when we treated with TDZ. Further research is needed to define the relationship between leaf abscission and stoma damage.

### 3.2. ROS Homeostasis in Leaf Abscission in Respond to TDZ

Many environmental stresses, such as drought, UV, chilling, salt, and pathogen attack and hormone treatment, can induce leaf abscission in plants [3,20,32,37]. ROS play an important role in plant growth and development, especially in stress tolerance [38]. In plants, ROS regulate many developmental processes such as cell proliferation and differentiation, seed germination, programmed cell death, senescence [39,40,41,42,43]. ROS include H_2_O_2_, O^2−^, singlet oxygen, and the hydroxyl radical [13,44]. A previous study has founds that excessive ROS accumulation causes DNA injury and damage, and further causes the apoptosis of leaf cells [36]. Deficiency of membrane integrity under stress is generally linked with excessive ROS, including H_2_O_2_ and O^2−^ and the increasing of the MDA level [43]. MDA content is supposed to an indicator of membrane integrity and injury for estimating cell damage under a variety of disadvantage conditions [45]. In this study, we show that O^2−^, H_2_O_2_, and MDA content all increased, especially H_2_O_2_ and MDA content. Therefore, leaf stomata damage maybe relate to ROS excessive accumulation and extreme increase of MDA content. RNA-seq analyses revealed an oxidation-reduction process (especially glutathione metabolism) participant in this process. In leaf abscission, H_2_O_2_ also plays an important role related to ethylene both in vitro and in vivo [14]. Plants resist water-deficit stress by shedding leaves and ROS play an important regulatory role in the process of cassava leaf abscission under drought stress [19]. The involvement of oxidative processes leads to leaf abscission under chilling and light stresses [17]. The increase of ROS accumulation and MDA content alter superoxide dismutase (SOD) activity and catalase (CAT) activity, and damage photosynthesis and cell structure under stress conditions [45,46]. Moreover, a previous study suggested that the activated enzymatic antioxidant system contributed to ROS scavenging and inhibition of lipid peroxidation [47,48]. This study showed that POD and SOD activity increased at 48 h after TDZ treatment; however, CAT activity decrease. Anthocyanins are known to have antioxidant activities. Recent studies have revealed that ROS could trigger anthocyanin accumulation as a feedback regulation, and anthocyanins decreased the ROS level and ROS-generating stresses in maintaining photosynthetic capacity [49,50]. Our research finds that anthocyanin also increased, which may be as antioxidant substances to protect leaf cells from harm. This result indicates that the effect of ROS scavenging is weakened and ROS homeostasis is broken during leaf abscission induced by TDZ. Additionally, the ABR and AZR are positively correlated with MDA, H_2_O_2_, and O^2−^ by phylogenetic generalized least squares approach. Therefore, ROS may be related to cotton leaf abscission induced by TDZ.

### 3.3. Photosynthesis and Carbohydrate Metabolism in Leaf Abscission Induced by TDZ

Photosynthesis is a process used by plants to convert light into chemical energy that can later be stored in carbohydrate molecules, such as sugar, which are synthesized from carbon dioxide and water. Photosynthesis is important for maintaining plant growth and development. In this study, we reveal that photosynthesis is severely disrupted, and no photosynthesis occurs after 24 h with TDZ treatment. One possible reason for this response could be that excessive ROS production and leaf cell structure destruction affect photosynthesis. This is consistent with previous studies, which showed that ROS could exacerbate the adverse effects on leaf photosynthesis [51]. Previous studies have shown that photooxidative stress can be caused by photosynthesis-derived ROS [52]. Many abiotic and biotic stresses impair the photosynthesis metabolism, leading to the yield of crop reduction [53,54]. However, our finding is that TDZ application significantly and rapidly decreased the Pn, Gs, and Tr of cotton leaf, and all of those indexes values become approximately zero. RNA-seq analyses showed that almost all of photosynthesis-related genes downregulate when induced by TDZ. Correlation analyses also reveal that there is a positive correlation between photosynthetic indexes and indexes of defoliation (AZR and ABR) in cotton. Those results further support the hypothesis that cotton leaf abscission may be related to dramatic changes in photosynthesis.

During photosynthesis, leaves chlorophyll capture the energy from the sun to create sugary carbohydrates, which allows the plant to grow. As plants aging, chlorophyll content of leaves decreases [55]. GO enrichment analysis found that most of the genes related to photosynthesis, chlorophyll metabolism, and carbon fixation are downregulated during leaf abscission induced by TDZ. This is consistent with the findings of previous research that most of the genes related to photosynthesis, chlorophyll metabolism, and carbon fixation are downregulated during cotton leaf senescence [56]. On the other hand, senescence, which is the final stage of organ development, can also cause leaf abscission [57]. In plants, the pigment is located in Photosystems I and II within the thylakoid membrane of chloroplasts, meaning that chlorophyll is a membrane-bound pigment [58,59]. Chlorophyll in plants is divided into binding chlorophyll and free chlorophyll, and chlorophyll can only work when it binds to proteins, while free chlorophyll can cause photo oxidative damage to cells. To avoid photo-oxidative damage to cells caused by free chlorophyll, plants must degrade these substances quickly. It is worth noting that chlorophyll a, chlorophyll b, and total chlorophyll significantly increased at 48 h after treatment in this study. One possible reason for this response could be that the free chlorophyll increased and could not be timely degraded. Our result shows that the chlorophyll a and chlorophyll b ratio decreased more than control plants. Adjusting the chlorophyll a/b ratio is mainly achieved through the “chlorophyll cycle”, which plays a crucial role in plant senescence [60,61].

Carbohydrate accumulation in leaf is from photosynthesis. Depression in photosynthesis decreases the production of photosynthesis (carbohydrate molecules) in leaf [62]. Carbohydrate molecules such as starch, sucrose, and soluble sugar are detected and the content of them all obviously decreased. Previously, there is research that found that carbohydrate stress could induce longan fruits abscission through ROS accumulation [21]. Nutritional stress established by BA treatment can induce fruit abscission and may transmit signals via ROS, sugar, and hormones signaling crosstalk [31]. In a previous study, carbohydrates played an important role in signal delivery and material communication [63]. The previous study found that carbohydrate may be a biochemical signal controlling fruits abscission [22], and carbon starvation increases flower abscission [23]. Our results find that carbohydrate molecules obviously decrease and ROS also severely accumulate, which supports the idea that it is possible that both ROS and sugar signaling jointly regulate downstream abscission signaling, and then, abscission signaling is transferred to the leaf abscission zone.

### 3.4. The Potential Crosstalk among Phenotypic Characteristic, ROS, Photosynthesis, and Carbohydrate Metabolism in Leaf Abscission Induced by TDZ

Through a series of signal delivery, there is material communication among leaves, petioles cells and abscission zone, and the leaves of cotton abscise. Notably, we found the potential crosstalk used a phylogenetic generalized least squares approach to test hypothesized relationships among the index of phenotypic characteristics, ROS metabolism, photosynthetic processes, and carbohydrate metabolism. All of ROS metabolism, photosynthetic processes, and carbohydrate metabolism were involve in the early response during cotton leaf abscission. A bigger reason for concern is that those three processes not only shown significantly close relationships but were also related to leaf abscission and formation of the abscission zone. Further study on signal delivery and material communication between leaf and leaf abscission zone could enhance our understanding about the mechanism of leaf abscission in the future.

## 4. Materials and Methods

### 4.1. Materials and Experimental Design

The cultivar CRI 12 and CRI 49 were developed by the Institute of Cotton Research, Chinese Academy of Agricultural Sciences (ICR-CAAS), and have been commercially used for many years. These two cultivars are mostly planted in Xinjiang cotton-growing areas in China. From our previous screening study, we selected those two genotypes in this study (data unpublished). A pot experiment was conducted in a greenhouse in the Institute of Cotton Research of the Chinese Academy of Agricultural Sciences (CAAS), Anyang city, Henan Province, China. The pot was 12 cm in diameter, 10 cm high, and filled with nutrient soil (The Pindstrup Group Origin: LV: Latavia/EE: Estonia/DK: Denmark/RU: Russia) and vermiculite (1:1), with only one plant in each pot. The room temperature was kept at 28 ± 2 °C day/night, and the photoperiod is 12-h dark/12-h light with light intensity at 100–150 μE m^−2^ s^−1^. The relative humidity of the room was maintained at 50–60%.

At the eight- leaf stage, healthy and uniform seedlings (30 days after sowing) were selected. Two treatments were designed as follows: water and 100 mg L-1 TDZ, denoted as CK and TDZ; sixty-three pots are needed for one treatment and one plant per pot. The experiments were repeated three times. The fourth leaf was sampled at 24 and 48 h after treatment. The leaves were washed with distilled water and cut along the main vein, then immediately frozen in liquid nitrogen and stored in an ultra-low temperature freezer (−80 °C) for subsequent physiological data measurement. Leaf samples of CRI 12 at 48 h were used for RNA sequencing.

### 4.2. Abscission Rate and Formation Rate of Abscission Zone

Abscission rate (ABR) = (Lb-La)/Lb; formation rate of abscission zone (AZR) = (AZb-AZa)/AZb. Lb is the number of leaves before treatment, and La is the number of leaves at day = 1, 2, 3, 4, 5, 6, 7 after treatment; AZb is the number of abscission zone existing (AZb = Lb) before treatment; AZa is the number of abscission zone formation at day = 1, 2, 3, 4, 5, 6, 7 after treatment.

### 4.3. Scanning Electron Microscopy of Leaf

Scanning electron microscopy analyses were conducted according to Du, M.W. et al. [25] with little modification. For the electron microscopy observations, the blade of the 4th leaf was fixed in 2.5% (*v:v*) glutaraldehyde solution in 100 mM sodium phosphate buffer (pH 7.4) for 10 h at room temperature, then changed 2.5% (*v:v*) glutaraldehyde, rinsed three times in buffer, rinsed two times in ultrapure water and then dehydrated in ethanol through a series of increasing concentrations(30%, 50%, 70%, 80%, 95%, 100%). Sputter coated sections were then examined at different magnifications with Hitachi S-530 scanning electron microscope (Hitachi, Japan).

### 4.4. Enzymes Activities, O_2_^−^, MDA, and H_2_O_2_ Content Assays in the Leaf

At 24 and 48 h after spraying 100 mg L^−1^ TDZ at the eight-leaf stage, the content of the superoxide(O^2−^), hydrogen peroxide(H_2_O_2_) and MDA was detected. The enzyme activities of CAT, POD, and SOD were detected. Frozen leaf tissues (0.5 g) were ground in 10 mL of 50 mM phosphate buffer (PSB) at pH 7.8 containing 1% polyvinylpyrrolidone (PVP), 0.2 mM EDTA-Na_2_, and 10 mM MgCl_2_. The homogenate was kept for 10 min under 4 °C and then centrifuged for 25 min with 15,000× *g* at 4 °C. The supernatant was used for measuring SOD, POD, CAT enzyme activities, O^2-^ generation rate, and soluble protein content [64,65,66]. Enzyme activity units were defined according to Wang [67].

Malondialdehyde (MDA) was extracted and detected as described by Wang [67], with slight modification. About 0.3 g leaf samples were ground in 8 mL of 10% (*w/v*) trichloroacetic acid (TCA), and the homogenate was centrifuged at 4000× *g* for 10 min. We mixed the supernatant (2 mL) with 0.6% (*w/v*) thiobarbituric acid (2 mL), and then heated the mixture in a 100 °C water bath for 15 min. The absorbance was measured at 450, 532, and 600 nm. The concentration of MDA was calculated by the following formula: C (μmol/L) = 6.45 × (OD _532_ − OD _600_) − 0.56 × OD 450. MDA content was expressed as μmol g^−1^ FW.

H_2_O_2_ concentration was measured by extracting 0.1 g fresh leaf samples in 0.1% TCA. After centrifugation at 12,000× *g* for 15 min, 0.5 mL of supernatant was mixed with 0.5 mL of 10 mM potassium phosphate buffer (pH 7.0) and 1 mL KI (1 M). Afterward, the OD value of solution mixture was read at 390 nm [68]. Finally, the hydrogen peroxide concentration was calculated using a standard curve of H_2_O_2_.

### 4.5. Photosynthetic Parameters, Carbohydrates and Soluble Protein Content

Net photosynthesis (Pn), stomatal conductance (Gs), intercellular CO_2_ concentration (Ci), and transpiration rate (Tr) of the fourth true leaf were measured at the eight-leaf stage using a portable photosynthesis system (Li-6800, Li-COR, Lincoln, NE, USA). Photosynthetic parameters were measured under 55% ± 5% relative humidity, 400 ± 5 μmol mol^−1^ CO_2_, and 9000 μmol m^−2^ s^−1^ photosynthetic.

Fresh leaf tissues (0.5 g) were ground by liquid nitrogen. The sucrose, starch, and fructose contents were determined by spectrophotometer analysis of ethanol-soluble extractions according to Hu [69], and starch content was extracted from ethanol-insoluble residue following previous described by Hu [70,71] and determined by spectrophotometer at OD_260_ using an anthrone reagent.

The soluble protein content was determined by Bradford reagent [72]. In addition, 0.1 mL extraction was shifted into a glass tube. Then, 5 mL coomassie brilliant blue G-250 solution was added to the glass tubes, then shaking the mixtures. The absorbance at 595 nm was measured. Bovine serum albumin was used to make a standard curve.

### 4.6. Measurement of Anthocyanin and Chlorophyll Contents

Anthocyanin was measured according to Jeong [73], and we defined OD _530_-0.25 OD _657_ as a unit (u). Chlorophyll contents were measured according to Pérez-Patricio. Approximately 0.05 g fresh leaf tissue samples were extracted with 95% ethanol. All samples were analyzed by a spectrophotometer (UV-1280, Shimadzu, Kyoto, Japan) at 470, 649, and 665 nm wave lengths, respectively. The chlorophyll a, b, total chlorophyll (Chla + Chlb), and carotenoid (Cxc) were computed using the following equations: (1) Chla (mg L^−1^) = (13.95 × A_665_) − (6.88× A_649_), (2) Chlb (mg L^−1^) = (24.96 × A_649_) − (7.32 × A_665_), and (3) Cxc (mg L^−1^) = (1000 × A_470_ − 2.05 × Chla − 114.8 × Chlb)/245.

### 4.7. RNA Extraction, cDNA Library Construction, and Sequencing

Trizol reagent (Invitrogen, CA, USA) was used to extract total RNA following the manufacturer’s procedure. The total RNA quantity and purity were detected with RIN (RNA integrity number) >7.0 using a Bioanalyzer 2100 and an RNA 1000 Nano LabChip Kit (Agilent, CA, USA). Then, the poly (A) RNA was obtained through two rounds of purification by purifying total RNA using poly-T oligo-attached magnetic beads. Following purification, the mRNA was fragmented into small pieces under elevated temperature. Then, the cDNA library was constructed by reverse-transcribed cleaved RNA fragments in accordance with the protocol for the mRNA Seq sample preparation kit (Illumina, San Diego, CA, USA). The average insert size for the paired-end libraries was 300 bp (±50 bp). The paired-end sequencing was performed using Illumina Novaseq™ 6000 (LC Sceiences, USA) by following the vendor’s recommended protocol, generating a total of 318.1 M raw paired-end reads.

Afterwards, we deeply filtered raw paired-end reads by removing the low-quality reads, reads containing sequencing adaptors, reads containing sequencing primer, and nucleotides with a q-quality score lower than 20. After that, a total of 278.4 M clean reads were obtained. The raw sequence data have been submitted to the NCBI Short Read Archive (SRA) with accession number PRJNA622598.

Clean reads of RNA-seq data were mapped to the *Gossypium hirsutum* reference genome (HAUv1, https://www.cottongen.org) using the HISAT package. HISAT allows multiple alignments per read (up to 20 by default) and a maximum of two mismatches when mapping the reads to the reference. HISAT builds a database of potential splice junctions and confirms these by comparing the previously unmapped reads against the database of putative junctions.

Then, the edgeR package was used to estimate the expression levels of all transcripts by fragments per kilobase of transcript per million fragments mapped (FPKM). The differentially expressed mRNAs and genes were selected with log2 (fold change) > 1 or log2 (fold change) < −1 and with statistical significance (*p*-value < 0.05) by R package.

### 4.8. Statistical Analyses

The data presented graphically are means (± SE) of at least three replications. The effect of TDZ treatment and water treatment was assessed using a one-way analysis of variance with the SAS statistic package (SAS 9.2, USA). Figures were drawn using the R and Excel software program (Microsoft Excel 2010).

## 5. Conclusions

In conclusion, after TDZ treatment, the ROS level increases, the structure of leaves are destroyed, photosynthesis is inhibited, and the carbon metabolism is unbalanced. ROS and carbon stress of leaves may jointly promote downstream shedding signals and regulate cotton leaf shedding. This work connects the cotton leaf abscission to ROS, photosynthesis, and carbohydrate metabolism induced by TDZ, and lays the groundwork for physiological and molecular change in cotton, which may be useful for future chemical defoliant research. In conclusion, our research will be beneficial for understanding the mechanism of cotton leaf abscission induced by TDZ.

## Figures and Tables

**Figure 1 ijms-21-02738-f001:**
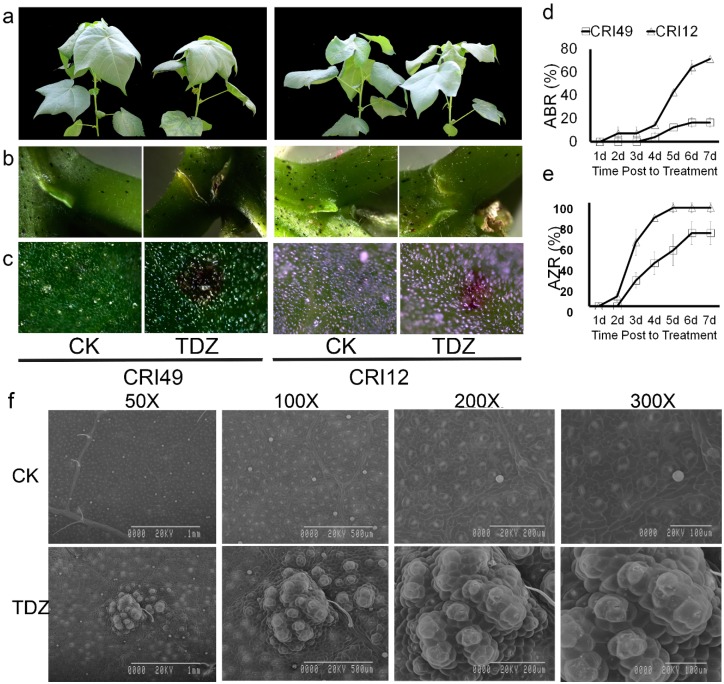
Phenotypic changes, abscission rate (ABR), and abscission zone formation rate (AZR) in CRI 49 and CRI 12, with or without Thidiazuron (TDZ) treatments. (**a**) whole plant (camera image); (**b**,**c**) abscission zone and leaf (T microscope image); (**d**,**e**) ABR and AZR represent abscission rate and abscission zone formation rate, respectively; (**f**) scanning electron microscopy of leaf epidermis cells of CRI 12. CK and TDZ represent control (water treatment) and 100 mg L^−1^ TDZ treatment, respectively.

**Figure 2 ijms-21-02738-f002:**
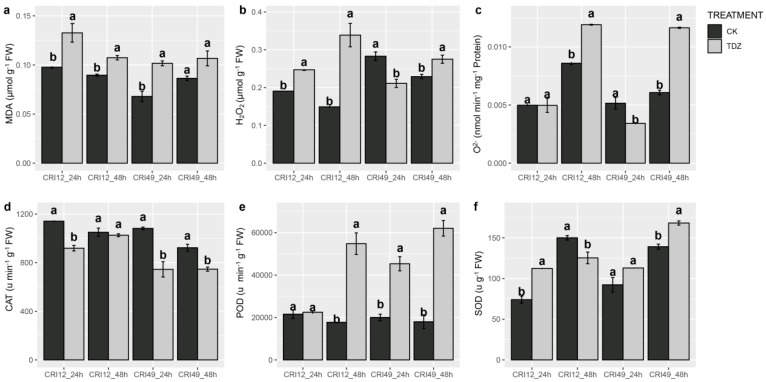
Effect of TDZ on malondialdehyde (MDA), H_2_O_2_ and O_2_^−^ content, and the activities of antioxidant enzymes in cotton leaves. (**a**) MDA content; (**b**) H_2_O_2_ content; (**c**) O_2_^−^ ptoduction rate; (**d**) CAT enzyme activity; (**e**) POD enzyme activity; (**f**) SOD enzyme activity. CK and TDZ represent control (water treatment) and 100 mg L-1 TDZ treatment, respectively. Values not sharing a common letter within the same column are significantly different (*p* < 0.05) according to Duncan’s test using SAS software.

**Figure 3 ijms-21-02738-f003:**
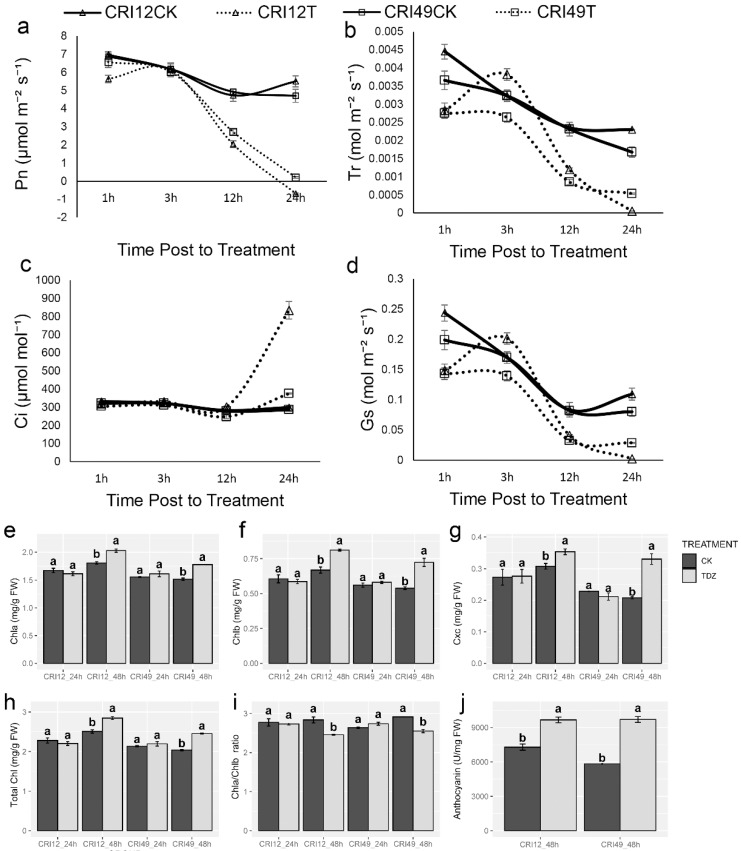
Photosynthetic parameters changes under TDZ treatment. (**a**) Net photosynthetic rate; (**b**) Transpiration rate; (**c**) Intercellular carbon dioxide concentration; (**d**) Stomatal conductance; (**e**) The content of chlorophyll a; (**f**) The content of chlorophyll b; (**g**) The content of carotenoid; (**h**) The content of total chlorophyll; (**i**) The ratio of chlorophyll a and chlorophyll b; (**j**) The content of anthocyanin. CRI12CK, CRI12T, CRI49CK, and CRI49T represent CK and TDZ treatment of CRI 12, and CRI 49, respectively. Values not sharing a common letter within the same column are significantly different (*p* < 0.05) according to Duncan’s test using SAS software.

**Figure 4 ijms-21-02738-f004:**
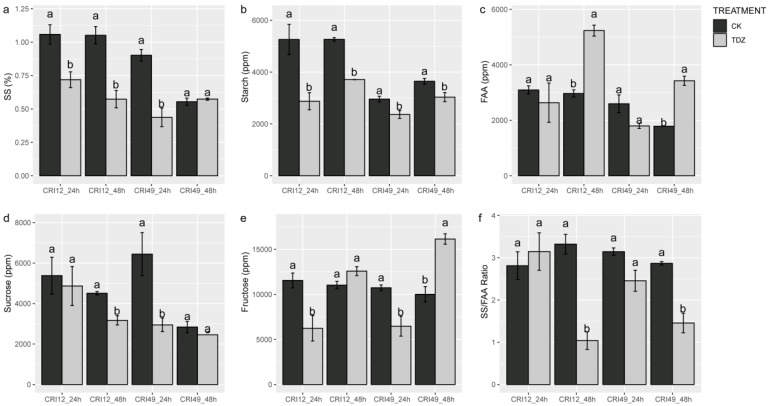
Soluble sugar (SS), sucrose, starch, fructose, and free amino acid (FAA) contents of leaf under TDZ treatment for 24 and 48 h. (**a**) The content of soluble sugar; (**b**) The content of starch; (**c**) The content of free amino acid; (**d**) The content of sucrose; (**e**) The content of fructose; (**f**) The ratio of soluble sugar and free amino acis. CK and TDZ represent control (without TDZ treatment) and 100 mg L^−1^ TDZ treatment, respectively. Values not sharing a common letter within the same column are significantly different (*p* < 0.05) according to Duncan’s test using SAS software.

**Figure 5 ijms-21-02738-f005:**
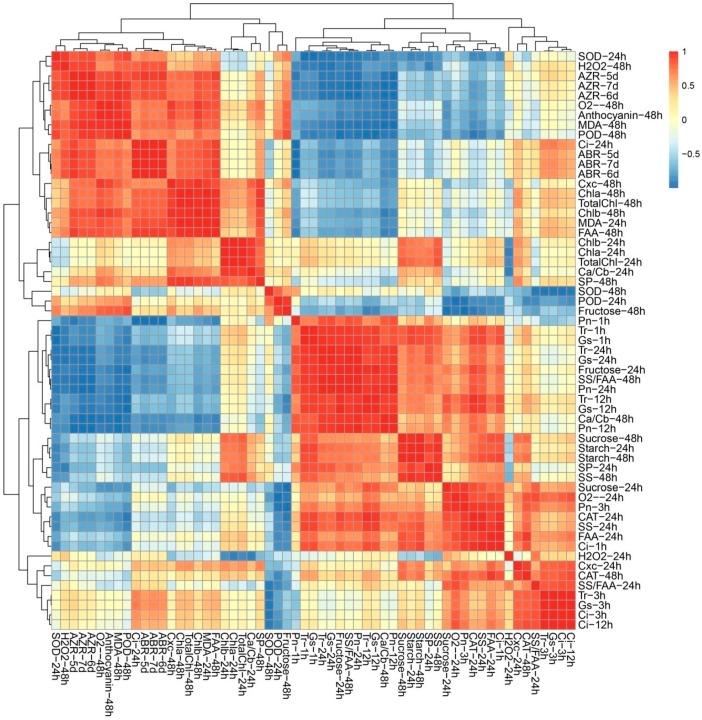
Correlation analyses among different indices in leaf.

**Figure 6 ijms-21-02738-f006:**
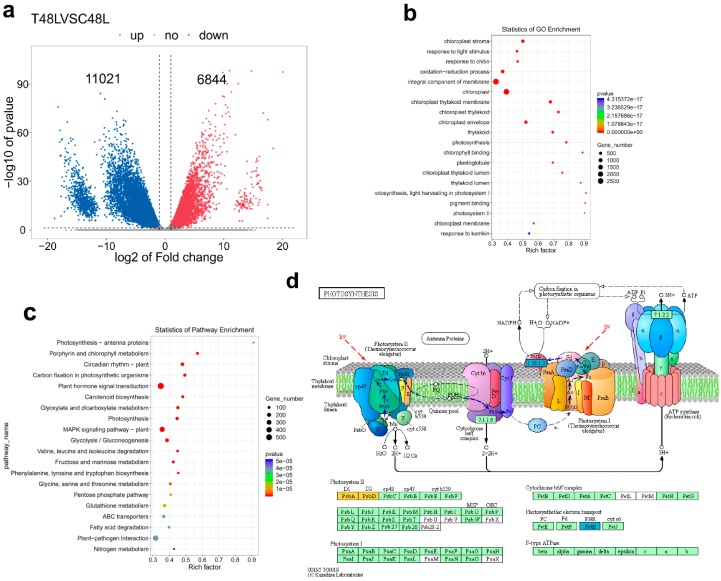
Function annotation of cotton differentially expressed genes (DEGs). (**a**) Venn diagram shows DEGs. Blue and red colors indicate down- and upregulated genes, respectively. (**b**) Gene ontology (GO) classification of the assembled cotton genes (Top 20 enrichment GO term). The colors represent the *p*-value level and the value gradually increases from blue to red. The solid circle size means the number of DEGs. The rich factor is bigger, the enrichment degree is higher. (**c**) Kyoto Encyclopaedia of Genes and Genomes (KEGG) pathway analyses (Top 20 enrichment KEGG pathway). The colors represent the *p*-value level and the value gradually increases from blue to red. The solid circle size means the number of DEGs. The rich factor is bigger, and the enrichment degree is correspondingly higher. (**d**) Photosynthesis related gene down-regulate induced by TDZ. Red and blue represent the significantly up- and downregulated transcription of notes to a KO node, respectively; orange represents the significantly both upregulated and downregulated transcription of notes to a ko node.

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
