# Peer review of "Chemical Defoliant Promotes Leaf Abscission by Altering ROS Metabolism and Photosynthetic Efficiency in Gossypium hirsutum"

_ijms, 2020, doi:10.3390/ijms21082738_

Round 1
Reviewer 1 Report
The paper is generally well written and covers integral details concerning plant physiology of leaf defoliation. Following are editing comments:
- Line 43-44. Sentences are difficult to follow, need to be revised.
- Line 50. Delete “These”
- Line 55, 56, 59, 61 – add scientific names.
- Line 69. Defoliants can also promote boll opening.
- Line 75-76. Problems with defoliants are usually associated with timing and rate of application, and interactions with temperature. These should be mentioned as “some problems”.
- Line 78 and 81. What is meant by “cross talk” and “crosstalk”?
- Line 88. It is highly unusual to present “2. Results” before “3. Discussion” before “4. Materials and methods. Results and discussion cannot be reviewed with first reviewing materials and methods.
- Line 161-172 – I did not see Figure 5 cited.
- Line 334. you provide any description of CRI-12 and CRI-49? If released cultivars, they should have published information
- Line 337. Describe nutrient soil.
- Line 341. How much time to reach 8-leaf stage? How many pots/plants per treatment?
Author Response
Response to Reviewer 1 Comments
Point 1:Line 43-44. Sentences are difficult to follow, need to be revised.
Response 1: Line 43. Thanks for your kind suggestion. I have revised those sentences. The new sentences are given below. “Previous studies have found that ethylene accelerates organ abscission, while auxin inhibits this process.”
Point 2:Line 50. Delete “These”
Response 2: Line 50-51. I have deleted the word “These”.
Point 3:Line 55, 56, 59, 61 – add scientific names.
Response 3: Line 55, 57, 60, 62. I have added the crop`s scientific names according to the suggestion. cassava (Manihot esculenta Crantz), olives (Olea europaea L.), longan (Dimocarpus longan Lour.), Citrus (Citrus unshiu [Mak.] Marc.).
Point 4:Line 69. Defoliants can also promote boll opening.
Response 4: Line 70-73. Thanks for your constructive suggestion. I have add this idea in the below sentence. “Applying chemical defoliants before harvesting can promote the shedding of cotton leaves as well as promote boll opening and thus effectively reduce the content of impurities in raw cotton and enhance the harvest efficiency, especially for the mechanically harvested cotton.”
Point 5:Line 75-76. Problems with defoliants are usually associated with timing and rate of application, and interactions with temperature. These should be mentioned as “some problems”.
Response 5: Line 76-79. Thanks, this is a good recommendation. I have adjusted the sentence and added some problems associated with defoliants. The new sentences are given below.
“In addition, there are still some problems in the practical use of defoliant, for example, the leaves dehydrated but not shed, and the effect of defoliants was tightly associated with the application schedule, dosage and air temperature.”
Point 6:Line 78 and 81. What is meant by “cross talk” and “crosstalk”?
Response 6: Line 81. I changed the spell of “cross talk” to “crosstalk”.
Point 7:Line 88. It is highly unusual to present “2. Results” before “3. Discussion” before “4. Materials and methods. Results and discussion cannot be reviewed with first reviewing materials and methods.
Response 7: Line 91. This is the requirement of IJMS journals.
Point 8:Line 161-172 – I did not see Figure 5 cited.
Response 8: Line 173-176. Thank you for reminding me to add graphics sequence “Figure 5”.
Point 9:Line 334. you provide any description of CRI-12 and CRI-49? If released cultivars, they should have published information
Response 9: Line 349-354. Detailed information of CRI 12 and CRI 49 were added in the manuscript according to the suggestion.
“The cultivar CRI 12 and CRI 49 were developed by the Institute of Cotton Research, Chinese Academy of Agricultural Sciences (ICR-CAAS) and have been commercially used for many years. These two cultivars are mostly planted in Xinjiang cotton growing areas in China. From our previous screening study, we selected those two genotypes in this study (Data unpublished).”
Point 10:Line 337. Describe nutrient soil.
Response 10: Line 353-354. I have described the origin of nutrient soil.
Point 11:Line 341. How much time to reach 8-leaf stage? How many pots/plants per treatment?
Response 11: Line 358-360. In this study, it is about 30 days to reach 8-leaf stage. 63 pots are needed for one treatment and one plant per pot.
Reviewer 2 Report
Fig1f, the last 2 pictures on 200x and 300x are switched. I think it is better to take more pictures at different stages during leaf abscission to provide detailed information.
line122, 124, full name of MDA, CAT, POD, SOD.
Fig2,3 and 4, p value of t-test is required for significant changes.
Extensive discussion regarding the relation between ROS/carbohydrate and abscission is required, linked with published works. Carbohydrate accumulation is important for organ abscission in many plants, removing of leaf is well-known to increase pedicel abscission in tomato. Agree with this, photosynthesis is affected by TDZ treatment.
Author Response
Response to Reviewer 2 Comments
Point 1: Fig1f, the last 2 pictures on 200x and 300x are switched. I think it is better to take more pictures at different stages during leaf abscission to provide detailed information.
Response 1: Line 113. Thanks for reminding me about it. I have switched the picture and placed it to the correct location.
Thanks for your suggestion again. In the present study, we considered only 2 days after treatment for Scanning electron microscopy of leaf epidermis cells. It is a good suggestion. In the future study, we would take more pictures at different stages during leaf abscission.
Point 2:line122, 124, full name of MDA, CAT, POD, SOD.
Response 2: Line 125-128. In the manuscript, I have added the full name of MDA, CAT, POD, SOD. “MDA (malondialdehyde), CAT (catalase), POD (peroxidase) and SOD (superoxide dismutase)”
Point 3:Fig2,3 and 4, p value of t-test is required for significant changes.
Response 3: Line 133-134,150-152,167-168. I have added the p value in the legend. The description of p value is finished as seen below. “Values not sharing a common letter within the same column are significantly different (P< 0.05) according to Duncan’s test using SAS software.”
Point 4:Extensive discussion regarding the relation between ROS/carbohydrate and abscission is required, linked with published works. Carbohydrate accumulation is important for organ abscission in many plants, removing of leaf is well-known to increase pedicel abscission in tomato. Agree with this, photosynthesis is affected by TDZ treatment.
Response 4: Line 268-271,329-331. Thank you for constructive suggestion and agreement. The discussion of between ROS/carbohydrate and abscission is really important and required. I have added this part and discussed their relationship in our manuscript now.